# Mitochondria as a Cellular Hub in Infection and Inflammation

**DOI:** 10.3390/ijms222111338

**Published:** 2021-10-20

**Authors:** Pauline Andrieux, Christophe Chevillard, Edecio Cunha-Neto, João Paulo Silva Nunes

**Affiliations:** 1Inserm, INSERM, UMR_1090, Aix Marseille Université, TAGC Theories and Approaches of Genomic Complexity, Institut MarMaRa, 13288 Marseille, France; Pauline.Andrieux@outlook.com; 2Laboratory of Immunology, Heart Institute (InCor), Division of Clinical Immunology and Allergy, School of Medicine, University of São Paulo, São Paulo 05403-000, Brazil; edecunha@gmail.com; 3Institute for Investigation in Immunology (iii), INCT, São Paulo 05403-000, Brazil

**Keywords:** mitochondria, inflammation, infection, mitochondria dysfunction, mitochondrial bioenergetics, infection disease, inflammatory disease

## Abstract

Mitochondria are the energy center of the cell. They are found in the cell cytoplasm as dynamic networks where they adapt energy production based on the cell’s needs. They are also at the center of the proinflammatory response and have essential roles in the response against pathogenic infections. Mitochondria are a major site for production of Reactive Oxygen Species (ROS; or free radicals), which are essential to fight infection. However, excessive and uncontrolled production can become deleterious to the cell, leading to mitochondrial and tissue damage. Pathogens exploit the role of mitochondria during infection by affecting the oxidative phosphorylation mechanism (OXPHOS), mitochondrial network and disrupting the communication between the nucleus and the mitochondria. The role of mitochondria in these biological processes makes these organelle good targets for the development of therapeutic strategies. In this review, we presented a summary of the endosymbiotic origin of mitochondria and their involvement in the pathogen response, as well as the potential promising mitochondrial targets for the fight against infectious diseases and chronic inflammatory diseases.

## 1. Introduction

Mitochondria are membrane-bound organelles, essential to produce energy in the form of ATP, because the phosphate is a high-energy bond and provides energy for other reactions within the cell. In addition to producing energy, mitochondria store calcium for cell signaling activities, generate heat and mediate cell growth and death. The number of mitochondria per cell varies widely according to the local energy demand. However, restricting mitochondria merely as an energy power organelle is to miss the big picture. The mitochondria’s oxidative phosphorylation (OXPHOS) is a major site for the production of reactive oxygen species (ROS; or free radicals) due to the high propensity for aberrant release of free electrons. While several different antioxidant proteins within the mitochondria scavenge and neutralize these molecules, excessive ROS production may inflict damage to mtDNA. Mitochondrial dynamics and energetics play a central role in proinflammatory signaling, thus are essential organelles for the control of innate immunity and inflammatory response against infectious pathogens.

### An Evolutionary Cooperation between Mitochondria and Eukaryotic Cells

The eukaryotic term includes all multicellular organisms (such as animals, fungi and plants) and unicellular organisms (e.g., protozoa) composed of eukaryotic cells. The eukaryotic cell is defined essentially by the existence of a structured nucleus and the presence of organelles in the cytoplasm such as mitochondria in contrast to prokaryotes. The mitochondria are the powerhouse of the cell, indeed, through the OXPHOS mechanism. The first descriptions of these organelles were performed in 1890 by Richard Altmann. He observed the existence of granula inside the cells stained with osmium and called them “bioblasts”. He described them as “elementary organisms” with autonomy in terms of metabolism and genetics. The name mitochondria came only later, in 1898, when Carl Benda named it from the Greek “mitos” (thread) and “chondros” (granule). Similarities between mitochondria and prokaryotes can also be found in the composition of their membranes. Indeed, both the inner membrane of the mitochondria and prokaryotic membrane share a phospholipid essential to the organization of these membranes: cardiolipin [1]. Moreover, genetic information is stored in the form of circular DNA in prokaryotes and mitochondria [2].

These similarities raised some questions about the origin of the eukaryotic cell. Previously, it was considered that the eukaryotic cells descended from an ancestral lineage that gradually became more complex due to numerous mutations. However, in 1918, Paul Poitier talked about symbionts: “These organelles would be for me nothing other than symbiotic bacteria, what I call symbionts”. Several years later, in 1967 the endosymbiotic theory was developed by Margulis “During the course of the evolution of mitosis, photosynthetic plastids (themselves derived from prokaryotes) were symbiotically acquired by some of these protozoans to form the eukaryotic algae and the green plants.” [3]. In 1998, Martin and Müller proposed the “hydrogen hypothesis for the first eukaryote”. This hypothesis suggested the mutualist symbiosis between an Archaean type host cell and an alphaproteobacteria. Indeed, the host cell would need to live on hydrogen in the medium and would produce methane in metabolic residues. Whereas the symbiont, alphaproteobacteria would be an optional aerobic organism that either carry out the fermentation that produces hydrogen as waste or breathe in the presence of oxygen [4]. Both the host and the symbiont benefit from this association.

After several studies to confirm the prokaryotic origin of the mitochondria, we could wonder when the evolution of the eukaryotic cell’s endosymbiosis has occurred. Several theories on the occurrence of mitochondria could explain the formation of prokaryotic cells. The first model (mito-late) is referred to by the event that the mitochondria appeared when the cell had already acquired a certain complexity. A second model suggested that mitochondria appeared before the complexification of the cell, and it would be precisely the acquisition of this organelle which allowed the cell to: the mito-early model.

Regardless of whether the mitochondria arrived early or late, we can think that their appearance led to a number of changes within the host that allowed its evolution to the eukaryotic cells we know today. Indeed, the evolution of the endomembrane system is part of the great changes brought by the endosymbiosis of an archaea and an alphaproteobacteria. Some biologists believed that the mitochondria within the archaea exerted a pressure that allowed the establishment of the endomembrane system, in particular through the secretion of vesicles [5].

The mitochondria are composed of a circular genome of 16.6 kb [1]. Unlike nuclear DNA, the mtDNA is not associated with histone proteins and is present in one or more copies per organelle. This genome encodes 13 proteins required for the OXPHOS, the 12S and 16S rRNAs and 22 tRNAs [6]. Due to the conservation of the primary structure of the different rRNA subunits, it was possible to elaborate the first phylogenetic tree between mitochondria and proteobacteria [7,8,9], and which made it possible to identify alphaproteobacteria as close to the mitochondria [10]. When comparing this genome with the alphaproteobacteria genome, although homologous, a large part of the mitochondrial genome was lost from its ancestral alphaproteobacteria [11,12]. The loss of these genes could be explained by the elimination of genes that were not essential for the correct functioning of this microorganism in its new environment. If genes with the same usefulness are found in the nucleus, we might think that those found in the mitochondria were lost to avoid costly energy redundancy [11,12,13,14]. This could also be explained by a gene transfer from the mitochondrial genome to the nuclear genome [15,16,17]. Finally, some studies suggest a suppression of redundant and non-essential genes by random mutations [18].

In addition, mitochondria also played a role in the mechanism of programmed cell death, apoptosis. Apoptosis is an essential mechanism of homeostasis [19], apoptotic signals lead to permeabilization of the mitochondrial membrane and release of cytochrome C, which activates caspases. This activation leads to chromatin condensation, nuclear fragmentation and ultimately the cells are fragmented and eliminated by macrophages without the induction of inflammation. Kroemer was the first to propose that apoptosis evolved during the domestication of ancestral mitochondria with the ancestral host cells as a defense mechanism against the host cell [20]. A more recent study, interested in reconstructing the apoptotic machinery at the level of the ancestral mitochondria, has shown that the establishment of apoptosis is the result of a co-evolution between the ancestral mitochondria and the host. In the course of evolution, the ancestral mitochondria and the host have each set up weapons to eliminate the other. It is the co-evolution of these two organisms that has allowed creating the programmed cell death mechanism as we know today [21]. The ancestral mitochondria had indeed released toxins and proteases such as caspases and meta caspases against the host. The host has evolved to protect itself by releasing protease inhibitors. These defense mechanisms have continued until it stabilized and gave rise to the programmed cell death mechanism [21].

The OXPHOS system is also an indicator of co-evolution between the ancestral mitochondria and the host cell since we found a reduction of the ancestral mitochondrial genome through the host cell genome profile. Indeed, oxidative phosphorylation protein components, which is an essential mechanism of this organelle, is encoded both by mtDNA coding factors and by the nuclear genome [22]. Knowing that the mechanism of oxidative phosphorylation is a mechanism inherited from the ancestral mitochondria, this is proof that during the domestication of the mitochondria by the cell, gene exchanges were carried out to favor the survival of the two symbionts. Moreover, the co-evolution between the ancestral mitochondria and the host was done in a very specific way for each species. Indeed, cellular hybrids made between a human nucleus and chimpanzee or gorilla mitochondria show a 20% decrease in endogenous cellular respiration [23,24]. The respiratory chain allows the generation of energy for the cell. For many, the acquisition of this mechanism has been a pillar of the evolution of eukaryotic cells. By increasing energy production [25], the ancestral cell was able to extend its genome and to allow an increase in cellular complexity [26]. Indeed, this increased energy allowed the eukaryotic cell to set up several mechanisms such as the cell cycle, endomembrane traffic [26], the nucleus compartmentalization [27] and the ability to become multicellular.

## 2. Mitochondria: General Concepts

### 2.1. Mitochondrial Respiratory Complexes

The mitochondria are composed of two lipid bilayer membranes, the outer and the inner membrane, which are separated by the intermembrane space. The inner membrane is organized in the form of invaginations called ridges. Under the inner membrane is located the mitochondrial matrix, in which the mitochondrial DNA and ribosomes are found (Figure 1).

The mitochondrial respiratory chain corresponds to a protein complex association at the level of the mitochondrial inner membrane. This respiratory oxidative phosphorylation chain is a key biological process that allows the phosphorylation of ADP to ATP through ATP synthase thanks to the energy released by the oxidation of electron donors (NADH and FADH2) through the respiratory chain [28].

The two electron donors NADH and FADH2 come from the tricarboxylic acid cycle (TCA cycle). Acetyl-coenzyme A (Acetyl-CoA) is a metabolite derived from the degradation of glucose, fatty acids and amino acids and is the first intermediate metabolite of the TCA pathway, being later combined with oxaloacetate to generate citrate. Citrate is converted into isocitrate, which is converted by two oxidative decarboxylations in α-ketoglutarate and succinyl-CoA. This reaction generates CO_2_ and two molecules of NADH. The NADH generated at this stage transfers electrons mainly to the complex I of the respiratory chain. After succinyl-CoA is converted into succinate, the oxidation of succinate generates fumarate and the electron donor FADH2, which will donor electrons to complex II. Later, the fumarate is converted into malate then in oxaloacetate and the cycle then restarts [29].

The respiratory OXPHOS chain is composed of five complexes and two coenzymes. Complex I is a NADH: coenzyme Q oxidoreductase, composed of 46 subunits, 7 of these subunits are encoded by mtDNA and the others by nuclear DNA. The 14 central subunits execute the bioenergetic function of this complex. In addition to these central subunits the complex I may have accessory subunits, but these subunits are not implicated in bioenergetic function, their function is not well known, however it is possible that they could improve electron transport [30].

Complex I allows the oxidation of NADH by the quinone coenzyme Q10 (CoQ10), releasing two electrons to Complex III and pumping four protons (H^+^) from the mitochondrial matrix to the inter-membrane space.

Complex II is exclusively encoded by the nuclear DNA and has four subunits with a succinate: coenzyme Q reductase mechanism of action. Its main role in the OXPHOS is to transfer the electrons from the succinate to fumarate oxidation to CoQ10. Since this redox reaction is of low energy, complex II does not contribute to the H^+^ gradient in the mitochondria.

Complex III is a CoQ10-cytochrome C oxidoreductase composed of 10 subunits. The cytochrome B, a central protein of this complex, is encoded by mtDNA and the other subunits are encoded by nuclear DNA. The passage of electrons from CoQ10 to cytochrome C pumps four H^+^ to the intermembrane space.

Cytochrome C enables the transition of electrons between the complexes III and IV. The complex IV has a cytochrome C oxidase action which transports two protons in the opposite direction of the gradient and reduces O_2_ in a H_2_O molecule. This complex is composed of 20 subunits, 3 encoded by mtDNA. Finally, all the protons present in intermembrane spaces pass through the ATP synthase. The force of this proton gradient leads to the synthesis of ATP from ADP and inorganic phosphate (Figure 2) [28].

The ATP synthase is composed of 29 subunits, these subunits can be separated in two functionally regions: F_0_ and F_1_. F_0_ is in the intermembrane and F_1_ is on the mitochondrial matrix. Throughout all major respiratory chain complexes, the passage of electrons pumps protons outwards. This flow creates an electrochemical energy inside the mitochondria, called the proton motive force [31].

The mitochondrial complexes stably interact with each other in the supercomplexes as proposed by the “Solid-state model” [32]. Two different supercomplexes were isolated from bovine heart and yeast cells using a blue native polyacrylamide gel electrophoresis (BN-PAGE) technique: I_1_III_2_IV_4_ and III_2_IV_4_. The association of complexes I, III and IV is called “respirasome” (Figure 3) [33,34].

In order to study the direct passage of electrons between the complexes, the metabolic flux control coefficient was calculated [35] and an association between complexes I and III was observed, while complex IV seems to be randomly distributed. In addition, the authors did not identify a direct electron channel between complex II and complexes III and IV [36]. In mammals, super complexes are most often composed of complexes I, III and IV, in the form I + III_2_ + IV_1__–4_ and I + III_2_. It has been shown that there is an increase in these complexes in the human skeletal muscles during effort, which shows that these super complexes are encoded according to the energy demand [37].

Another argument in favor of solid-state models is that the stability of supercomplexes depends on the complexes that constitute them. Indeed, a mutation in the assembly site in complex III leads to a deficiency of complex III as well as supercomplex I + III and a reduction of the amount of complex I. [38]. Complex IV is also necessary for the assembly or stability of complex I [39].

The organization of the mitochondrial ridges also participate in the stability of the supercomplexes. The mitochondrial ridges owe their organization on one side to the mitochondrial contact site and cristae organizing system (MICOS), this complex is found at the base of the ridges [40,41,42]. The MIC10 subunit of the MICOS complex is particularly important in the formation of these ridges [43,44]. ATP synthase also plays an important role in the stabilization of these ridges, being organized in dimers at the top of the ridges [45,46]. The other respiratory complexes seem to be located at the level of the flat surfaces of the ridges [47,48].

### 2.2. Mitochondrial Dynamics

Mitochondrial fission and fusion mechanisms maintain functional mitochondria when the cells experience metabolic or environmental stresses (Figure 4).

Fusion helps mitigate stress by mixing the contents of partially damaged with fully functional mitochondria as a form of complementation. Fission is needed to create new mitochondria, but it also contributes to quality control by enabling the removal of damaged mitochondria and can facilitate apoptosis during high levels of cellular stress. Mitochondrial fission and fusion processes are both mediated by large guanosine triphosphatases (GTPases). Fission is mediated by dynamin-related protein 1 (DRP1), which is recruited from the cytosol to form spirals around mitochondria that constrict to sever both inner and outer membranes. Fusion between mitochondrial outer membranes is mediated by membrane-anchored dynamin family members named mitofusins MFN1 and MFN2, whereas fusion between mitochondrial inner membranes is mediated by a single dynamin family member called OPA1 mitochondrial dynamin-like GTPase (OPA1) in mammals. Mitochondrial fission and fusion machineries are regulated by proteolysis and posttranslational modifications [49]. The remodeling of the ridges by the ablation of OPA1 would lead to a disturbance in the formation of super complexes and a decrease in respiration [50].

### 2.3. Mitophagy

Mitophagy is an essential mechanism in the homeostasis of the mitochondrial network because it mediates the removal of defective mitochondria. Two proteins are known to initiate this process: PTEN-induced putative kinase 1 (PINK1) and Parkin [51,52]. Under non-dysfunctional conditions, the PINK1 protein, located at the mitochondrial membrane, is degraded by two proteases, the matrix processing peptidases (MPP) and the presenilin-associated rhomboid like (PARL). However, infection or inflammatory processes can promote depolarization of the mitochondrial membrane, a process known to inhibit the cleavage of PINK1, which leads to the accumulation of PINK1 at the mitochondrial membrane. PINK1-rich membrane promotes activation of Parkin by phosphorylating SER65 in the ubiquitin-like (UBL) domain, promoting parkin-dependent ubiquitination of substrates in the mitochondria [52]. Additionally, PINK1 also phosphorylates ubiquitin monomers at SER65 in the cytosol [53,54,55], promoting arrangement of ubiquitin chains that will mediate ubiquitination of mitochondrial proteins. Mitochondrial proteins that have been ubiquitinated will bind to autophagy receptors such as optineurin and NDP52 recruited by PINK1 [56] that bind to the microtubule-associated protein 1A/1B-light chain 3 (LC3) protein linked to the phagophore. This will create the mitophagosome, which fuses with the lysosome to destroy the damaged mitochondria [57].

## 3. Role of Mitochondria in Inflammation and Infection

In different ways, mitochondria play a central role in the innate immune response. It is at the center of the inflammatory response in the case of a viral or bacterial infection or in cellular damage. Indeed, in the case of an infection by a pathogenic agent, the microorganisms will be detected by pattern-recognition receptors (PRR) that recognize pathogen-associated molecular patterns (PAMPs), such as flagellins, lipopolysaccharide, mannose, nucleic acids and proteins and the danger-associated molecular motifs (DAMPs) molecules derived from damaged cells pathway (Figure 5).

There are four families of PRRs, toll-like receptors TLRs [58], (NOD)-like receptors (NLRs), C-type lectin receptors (CLRs) and retinoic acid-inducible gene I (RIG-I)-like receptors (RLRs). Each PRR allows the secretion of type I interferon and pro-inflammatory cytokines via NF-kB [59].

In the case of viral infection, the mitochondria play a role in the detection of the virus. Indeed, the MAVS (mitochondrial antiviral signaling protein), a protein located at the outer membrane of the mitochondria, plays a central role of the RLR receptor signaling pathway, which leads to the production of pro-inflammatory cytokines and type 1 interferon (IFN) [60,61,62,63]. In fact, two cytoplasmic PRR, the retinoic-acid-inducible gene (RIG-1) and melanoma differentiation-associated protein 5 (MDA5) recognize different PAMPS such as dsRNA viral or 5′-triphosphate (ppp) and single-stranded RNA (ssRNA) [64]. These PRR are regulated by DExH-Box Helicase 58 (DHX58) [65], where DHX58 positively regulates MDA5 and negatively regulates RIG-1 [66]. RIG-1 and MAD5 are cytosolic helicases with an ATPase activity. The C-terminal links with viral RNA and the N-terminal binds to two caspase recruitment domains (CARD) in tandem to link with MAVS protein [64,65,66,67,68]. After the interaction with DHX58, MAVS is activated and forms the MAVS signalosome by recruiting different proteins, such as TNF receptor associated factors TRAF3 and TRAF6 [69,70]. The recruitment of TRAF3 and TRAF6 induces expression of type I IFN and NF-ĸB pathway activation. Indeed, TRAF3 activates the TANK/NEMO/iKKe/TBK complex, which will phosphorylate IRF7 and IRF3 dimers, allowing their translocation into the nucleus and their binding to ISRE sequences at the promoters of type I IFN-regulated genes. On the other hand, TRAF6 activates the IKK complex. This complex activates NF-ĸB by phosphorylating its inhibitor IKBα. Once activated, NF-ĸB translocate to the nucleus to induce the expression of pro-inflammatory genes [71].

In addition, mitochondrial dynamics regulate the RLR signaling pathway. The mitochondria are a dynamic network that change in response to biological demands through fusion and fission. It was shown that interaction between MAVS and MFNs protein as well as mitochondrial fusion is necessary in the RLR signaling pathway [72]. Indeed, mitochondrial elongation and increased RLR response was observed after DRP1 and FIS1 inhibition by shRNA and siRNA in Hela cells infected by sendai virus (SeV) strain H4 [72]. Furthermore, knockout of MFN1 and MFN2 in mouse embryo fibroblasts (MEFs) reduced the RLR response against Sendai virus (SeV) and encephalomyocarditis virus (EMCV) [73]. So, disturbances of mitochondria fission and fusion affect the RLR-mediated response to virus.

The electron transport chain of the oxidative phosphorylation produces ROS, mainly in the complexes I and III, and to a lesser extent in complex II [74]. ROS are essential in antiviral signaling and antibacterial response. In the case of antiviral signaling, several observations suggest that increased cellular ROS amplifies the RLR response. It was shown that deletion of the autophagic vesicle formation gene ATG5 in mouse embryonic fibroblasts (MEFs) lead to an increase in ROS production and mitochondrial damage as well as increase in the RLR signaling [75,76]. In addition, authors described that production of type I interferon through RLR signaling is diminished by antioxidants the N-acetyl-L-cysteine (NAC) and propyl gallate (PG) [76]. These experiments show that constitutive levels of ROS are used by RLR for signaling. In addition, blockage of Complex I with rotenone in WT and ATG5 mutated cells increased mitochondrial ROS and RLR response against vesicular stomatitis virus infection (RIG-I stimulation) or Poly I:C transfection (MDA-5 stimulation).

The importance of ROS in the antibacterial response was shown via the ablation of regulatory protein Uncoupling Protein 2 (UCP2]. Studies have shown that electron leakage during ETC produces superoxides that diffuse to the internal membrane, oxidizing phospholipids and generating 4-hydroxynonenal [4HNE). The 4HNE activates the UCP that will uncouple the mitochondrial respiratory chain, therefore reducing the protonic force and the formation of superoxide [77]. Knockout of UCP2 in mice model was associated with higher mitochondrial ROS (mtROS) production, increased production of type I interferon and pro-inflammatory cytokines and thus increased antibacterial response [78,79]. LPS-stimulated macrophages had decreased UCP2 expression, which led to an increase in mtROS and higher production of pro-inflammatory cytokine and type I interferon [80]. Mitochondrial ROS is thought to participate in oxidative bursting in activated macrophages and these data revealed the important role of mitochondria-generated ROS in fighting infection.

Macrophages and dendritic cells eliminate microorganisms through phagocytosis. The proximity between phagosomes and mitochondria allows mtROS to cross the phagosome to eliminate the pathogen. After the activation of TLRs (TLR1, TLR2 and TLR4) by a pathogen, the increase of mtROS, as well as the approximation of mitochondria and phagosomes are done through the activation of the kinases Mst1 and Mst2. These kinases activate the GTPase Rac, which allows the assembly of the TRAF6-ECSIT complex [81]. Indeed, the TRAF6 factor translocates to the mitochondria and binds to ECSIT, playing an important role in complex I assembly. Complexing with TRACF6 will lead to an increase in mtROS production as well as the proximity of the mitochondria and the phagosome [82].

Mitochondria are also involved in an essential mechanism for fighting infection and cellular damage: the NRLP3 inflammasome. This mechanism allows the recruitment of inflammatory cells to the site of infection by allowing the secretion of IL-1b but also an induction of a TH1 response with an increase in cytotoxic activity with the secretion of IL-18 [83,84]. In addition to these two actions, the inflammasome is involved in a particular cell death: pyroptosis. This inflammatory cell death depends on the activation of the inflammasome and has the role of eliminating intracellular bacteria before they proliferate too much, as well as inducing a local inflammatory phenomenon with the recruitment of other immune cells [85,86,87].

The inflammasome is a multiprotein complex containing an adapter apoptosis-associated speck-like protein containing a C-terminal caspase recruitment domain (ASC), pro-caspase 1 and the specific member of the NOD-like receptor protein 3 (NLRP3) [88]. The activation of NLRP3 inflammasome is done on transcriptional and post translational levels. The first is related to activation of toll-like receptors (TLR) and production of pro-IL1b and pro-IL18 [89,90]. The second signal is based on the detection of PAMPs and danger-associated molecular motifs (DAMPs) that drive NLRP3 oligomerization and interaction with the ASC adaptor through the pyrin domain (PYD). Then, ASC recruits pro-caspase1 via the CARD domain. Once caspase 1 is activated, it cleaves the pro-IL1b and pro-IL18 which releases these inflammatory cytokines in circulation [88,91,92,93]. In addition, caspase 1 cleaves gasdermin D (Gsdmd), allowing the formation of plasma membrane pores and pyroptosis [94].

The mitochondria are very important in NLRP3 activation. Indeed, MAVS is necessary for the recruitment of NLRP3 in mitochondrial membranes [95]. The phospholipid cardiolipin translocates from the inner to the outer membrane of the mitochondria to bind to NLRP3 and promote its activation [96]. In addition, the mitochondrial ROS also participates in NLRP3 activation [97,98,99].

Infection by a pathogen is not the only cause of inflammation; indeed, tissue damage activates various inflammatory mechanisms through DAMPs. Since mitochondria have prokaryotic origins [4], it can be assumed that prokaryotic and mitochondrial components have similarities that may give them a high DAMP potential [100]. Indeed, a study has demonstrated the activation of the immune system by accumulation of mtDNA in the cytosol results in an antiviral immune response [101,102] and the oxidation of mtDNA leads inflammasome activation [101,103]. The cardiolipin is also considered a mitochondrial DAMP [104]. Moreover, as mentioned above, it allows the activation of the NLRP3 inflammasome [96].

In summary, the mitochondria are important to trigger the RLR, NLR or TLR-related innate immune response against virus, bacteria, fungi or sterile tissue damage.

## 4. Mitochondrial Bioenergetics and Dynamics in Infectious Diseases

The mitochondria play both bioenergetics and biosynthetic roles, since ATP and components of several macromolecules, such as amino acids, lipids and nucleotides, are generated in these organelle [105]. Mitochondria are also hubs for innate immune signaling against viruses and bacteria, thus involved in immunological response. Infectious pathogens have evolved mechanisms to target and modulate mitochondria to successfully replicate inside eukaryotic cells. The pathogen’s ability to exploit mitochondria is under close scrutiny over the past few years [106]. Considering the pleiotropic functions of mitochondria, it is not surprising that pathogens exploit the mitochondrial key roles during the infection affecting the oxidative phosphorylation complex and mitochondrial network. In this section, we briefly described how some pathogens affect mitochondria dynamics and energetics (Table 1).

### 4.1. Viruses

Viral infection is extremely dependent on the host cell machinery to successfully replicate and the mechanism by which viruses affect mitochondria is closely related to the virus type [164]. For example, infection with the alpha herpes virus Varicella zoster virus (VZV) was shown to induce persistent mitochondrial swelling and fragmentation throughout the time course of VZV infection, yet the mechanisms involved remain unknown [129]. Some studies suggest that VZV triggers the Warburg effect, i.e., metabolic shift from oxidative phosphorylation to glycolysis which regulates mitochondrial activity and thus promotes cell resistance to apoptosis and viral spread [129,130,131]. A similar study showed that alphaherpesviruses HSV-1 (herpes simplex virus type 1) and PRV (pseudorabies virus) also disrupt mitochondrial motility and morphology while promoting accumulation of intracellular Ca^2+^ in neurons, important mechanisms for viral growth and spread [107].

Infection with Epstein-Barr virus (EBV), an oncovirus associated with Burkitt lymphoma, reduces mitochondria content in differentiating monocytes [109], remodels B cell mitochondria by targeting the mitochondrial 1-Carbon pathway involved in the synthesis of purine, thymidylate and glutathione [110] and also induces mitochondria swelling, cyclophilin D dependent mitochondrial membrane permeabilization transition (MMPT), decreased mitochondrial membrane potential (Δψm) and ATP, increased ROS production and mitophagy and reduced apoptosis [111]. Additionally, proteins of human papillomavirus (HPV) interact with mitochondria, promoting detachment of mitochondria from microtubules, reducing or eliminating cristae compartment, resulting in decreased Δψm, increased production of mtROS, oxidative stress and DNA damage [126,127,128,165].

One of the most common effects in mitochondria caused by viruses is related to mitochondrial fragmentation. In order to suppress virus-induced apoptosis, hepatitis B (HBV) and hepatitis C (HCV) viruses modulate mitochondrial dynamics towards fission by the activation of dynamin-related protein 1 (DRP1) and mitochondrial fission factor (MFF), as well as increase in the expression of genes related to ubiquitin-proteasome system that mediates the targeting of proteins for degradation, such as LC3B, PARKIN and PINK1, thus activating mitophagy [112,113,166].

The infection with human immunodeficiency viruses HIV-1 also perturbs mitochondrial dynamics and bioenergetics. For example, mitochondria promote sustained calcium influx to support synaptic signaling and play an important role in the formation of the immunological synapse (IS), i.e., the nano-scale interface (gap) between T-cells [167]. The virological synapse of HIV-1 shares similar features with IS and the HIV-1 promotes mitochondria polarization and enhanced oxidative phosphorylation, thus facilitating HIV-1 replication and cell-to-cell transmission [114,115]. Moreover, the release of HIV proteins gp120 and Tat was shown to promote mitochondrial fragmentation by enhancing translocation of DRP1, decreased Δψm and perinuclear aggregation, generally followed by accumulation of ROS in the nucleus of human primary neurons [116,117].

While the aforementioned virus mediates mitochondrial fragmentation through activation of DRP1, dengue virus (DENGV), on the contrary, induces mitochondria elongation and reduced mitochondrial fragmentation via membrane-associated protein NS4B and suppression of DRP1 phosphorylation [108]. Similarly, infection with SARS-CoV and the novel SARS-CoV-2 virus was reported to induce mitochondria elongation through proteasomal degradation of DRP1 or decreased expression of mitochondrial fission promoting genes SOCS6 and MTFP1 [120,122,123,168]. SARS-CoV-2 spike protein peptides interacts with α7 nicotinic acetylcholine receptors (nAChRs) preventing mitochondria to release cytochrome C, avoiding mitochondria-driven apoptosis in a glioblastoma-derived cell line [169]. A study showed that COVID-19 patients had elevated mitochondrial ROS associated with tissue injury [124]. Moreover, the Influenza A virus and Influenza M2 virus induced mitochondrial elongation and fusion, production of mitochondrial ROS, driving innate immune inflammation which favors viral pathogenesis [118,119]. On the other hand, infection with the human rhinoviruses (HRV), which cause common cold and lower airways diseases, suppress mitochondrial ROS during infection to induce a transitory barrier-protective metabolic state that becomes exhausted as the infection progresses and leads to cellular damage [125]. I.e., while for some viruses, mitochondrial fragmentation and mitophagy are important mechanisms for viral replication because they favor cell survival and evasion of apoptosis, while for other viruses the maintenance of mitochondrial structure is essential to provide energetic provision to the infected cell before the virus replication cycle is complete [169].

### 4.2. Bacteria

In bacterial infections, host cells produce ROS as a defense mechanism to impair metabolism and growth of intracellular bacteria by inflicting damage to the pathogen’s lipids, proteins, and nucleic acids. However, some bacteria can benefit from ROS and thus sustain intracellular production of ROS to grow [170]. Mitochondrial dynamics play important roles in the pathogeny of bacteria. Mitochondrial fission, for instance, is an important element to eliminate infected cells and reduce cell-to-cell-spreading, thus modulation of apoptosis can support bacterial dissemination [140,171].

*Chlamydia trachomatis* is the most common sexually transmitted bacterium and can cause trachoma (blindness) and pneumoniae respiratory infection. A study showed that *C. trachomatis* indeed augments production of ROS through a mechanism that involves direct NOX-dependent ROS production and translocation of NLRX1 (member of the Nod-like receptor family) to mitochondria and for optimal chlamydial growth and replication [137,138]. Moreover, *C. trachomatis* requires mitochondrial ATP for normal development and consequently authors suggest that they preserve mitochondrial integrity through the microRNA miR-30c-5p-dependent inhibition of DRP1-mediated mitochondrial fission [136].

Infection with the diarrhea-causing bacteria *Shigella flexneri* was shown to drive mitochondria into fragmentation through DRP1 which allows the bacteria to evade the intracellular entrapment in cages-like structures composed of septins assembled by mitochondria [141]. Mitochondrial fission mediated by DRP1 is also an important preceding factor for vacuolating cytotoxin A (VacA) induced cell death of *Helicobacter pylori* infection [139]. Similarly, *Legionella pneumophilla*, which causes Legionnaire`s disease (an atypical pneumonia), controls mitochondrial dynamics to destabilize mitochondrial bioenergetics of infected cells via fission protein DRP1, causing mitochondrial fragmentation independent of cell death, dampening of mitochondrial respiration and promoting the Warburg-like phenotype in the infected cell that favors bacterial replication [135].

Conversely, some bacteria induce mitochondrial fragmentation independently of DRP1. Bacterial infection with *Lysteria monocytogenes* can cause severe sepsis, meningitis, or encephalitis and a study showed that this infection causes a transient Ca^2+^-dependent fragmentation of the mitochondrial networks through the secreted toxin listeriolysin O (LLO), impairing Δψm, mitochondrial respiration and decrease in intracellular content of ATP [133,134]. Moreover, *L. monocytogenes* increases the abundance of Mic10, a key component of the mitochondrial contact site and cristae organizing system (MICOS) in an LLO-dependent manner initiating mitochondrial fission and host cell infection [132].

Infection with *Mycobacterium tuberculosis* (Mtb), the causative agent of tuberculosis, has two phases, an early non-pathogenic phase and a pathogenic phase [172]. During the non-pathogenic phase, Mtb induces mitochondria fusion and robust activation of Δψm and ATP synthesis to inhibit apoptosis and sustain survival of the bacteria [143]. While the onset of the pathogenic phase is characterized by mitochondrial remodeling with induction of an irreversible MMPT following disruption of mitochondrial outer membrane (MOM), loss of Δψm, cytochrome C release and consequently necrosis [142,173].

### 4.3. Protozoa

Different from viruses and bacteria, protozoans carry their own mitochondria, and the mitochondrial dynamics are directly linked to the parasite life cycle. A recent review has documented the aspects of protozoan’s mitochondria and mitochondrion-related organelles dynamics [174]. Here, we summarized findings of how host’s mitochondria bioenergetics and dynamics are affected during protozoan infection.

Infection with the protozoan *Trypanosoma cruzi* can cause Chagas disease cardiomyopathy and megacolon [175,176]. Studies have shown that *T. cruzi* can elicit oxidative stress in host tissue by promoting increase in the production of cellular and mitochondrial ROS [148,149,150]. In addition, *T. cruzi* decreases the enzymatic activities of mitochondrial respiratory complexes [144,145], increased basal respiration, proton leak and ATP production [153]. Moreover, hearts of chronically *T.cruzi*-infected animals and Chagas disease patients have decreased activity of cytosolic glutathione peroxidase (GPx) and GSH and this was associated with decreased expression and activity of the mitochondrial antioxidant enzyme manganase superoxide dismutase (MnSOD) [146,147]. The combination of increased intracellular ROS and inadequate anti-oxidative response was described as crucial for *T. cruzi* survival, since *T. cruzi* takes advantage and uses ROS as “fuel” to grow, due to disturbed iron metabolism [177]. Mitochondria-associated damage has been observed in cardiomyocytes infected by *T. cruzi*, such as defects of the respiratory chain and enhanced oxidative stress [151,152]. A recent study showed that mice deficient for the mitochondrial folding protein CyPD, which is essential for mitochondrial permeability transition pore opening, had inhibited collapse of the Δψm and reduced severity of parasite aggression and spread of *T. cruzi* [178]. Recent studies revealed that *T. cruzi* tissue tropism is controlled by key metabolic pathways, such as glycolysis, fatty acid oxidation, acylcarnitine metabolism and amino acid catabolism [179,180]. Moreover, interactions between host-parasite-microbiome regulate the intensity of infection and progression of chagasic gastrointestinal disease [179]. *T. cruzi* can benefit from these pathways to growth and/or to evade antiparasitic immune response [180].

*Toxoplasma gondii*, the causative agent of toxoplasmosis, is known to cause encephalitis, retinitis and miscarriage [181] Data reported an increase of mitochondria fragmentation of human fibroblasts cells infected with the *T. gondii* and these mitochondria were recruited to the parasitophorous vacuole [158,182]. In addition, infected cells had metabolic perturbations of mitochondrial glycolysis [160] and OXPHOS complex [161], as well as increase of mitochondrial ROS, changes the amount of OXPHOS components and many other mitochondrial-related proteins [158,183]. In addition, host mitochondria fuse during *T. gondii* infection to limit fatty acids (FAs) uptake [159]. A proteomic study conducted by Blank and colleagues showed that *T. gondii* manipulation of host mitochondria requires association with key mitochondrial proteins, such as TOM70 and HSPA9 [184]. Another study showed that *T. gondii* unique dense granule antigen (GRA8) interacts with host proteins involved in mitochondria activation, ATP5A1-SIRT3 signaling pathways, thus contributing to a so-called mitochondrial metabolic resuscitation in order to maintain and regulate cellular metabolism and consequently parasite survival [166].

*Plasmodium* is a genus of protozoans that cause malaria, a highly inflammatory disease, being *Plasmodium falciparum* the deadliest in humans [185,186]. Shivappagowdar and co-workers provided new insights on malaria pathogenesis. They identified that the *P. falciparum* pore-forming proteins (PFPs) have damaging effects to primary vascular endothelial cells, where exposed cells had impaired Δψm, increased ROS and apoptosis [154]. In a similar approach, it was shown that infection by *Plasmodium yoelii* makes host hepatocytes susceptible to mitochondria-initiated apoptosis, through a mechanism that involve increased expression of the proapoptotic Bcl-2, leading to mitochondrial fission and apoptosis [155,156,157].

*Leishmania* spp. parasites cause cutaneous or visceral leishmaniosis, a neglected tropical disease that can cause skin ulcers, swelling of the spleen and liver. The infection with *Leishmania donovani* was reported to disrupt miRNA turnover in macrophages, causing mitochondrial depolarization through induced expression of the OXPHOS uncoupler uncoupling protein 2 (UCP2) [162]. This infection also affects the ER–mitochondria tethering mediated by mitofusin 2 [162]. Another study showed that macrophages infected with *L. infantum* switched from an early glycolytic metabolism to an oxidative phosphorylation, and this metabolic deviation requires SIRT1 and LKB1/AMPK [163].

Collectively, these studies revealed an intrinsic relationship between parasites and host mitochondria. In general, the infection triggers an initial protective effect to mitochondrial dynamics and host cell bioenergetics to sustain parasite proliferation and survival. Afterwards, the infection triggers a damaging response, by eliciting cell death and parasite release. To understand the different phases of infection and develop mitochondrial-target therapies is a promising strategy to ameliorate or cure infectious diseases.

## 5. Mito-Nuclear Crosstalk and Infection

Infection by pathogens changes the mitochondrial metabolic and oxidative profile of infected cells. One of the outcomes on mitochondria is fragmentation and excessive ROS production, which may lead to a decreased mitochondrial mass inside the host cell. The mitochondria cannot be made de novo and they are encoded by the both mitochondrial and nuclear DNA, thus the mito-nuclear communication is essential for the proper coordination of mitochondrial biogenesis. A dynamic regulatory system able to respond to the now challenging infectious environment is crucial [187]. Several studies already reported that dysregulation of the crosstalk between the nucleus and the mitochondria has an impact in the disease progression of different etiologies, such as cancer, diabetes and heart failure.

The mitonuclear regulation signaling can be both anterograde (nucleus to mitochondria) and retrograde (mitochondria to nucleus) and it is coordinated by transcription factors and coactivators that regulate the expression of both mitochondrial and nuclear genes in response to an infectious insult [188]. Metabolic or other damages that occur within mitochondria culminate in dysregulation of several nuclear genes through the retrograde signaling. The proteins belonging to the peroxisome-proliferator activated receptor coactivator-1 family (PGC1] are reported to be the master regulators of mitochondrial biogenesis, oxidative phosphorylation and response to oxidative stress in humans [189]. The amount of PGC-1a is closely correlated with the number of mitochondria and deficiency in PGC1a has been reported to cause mitochondrial dysfunction, metabolic derangements and cell death [190,191,192,193]. A recent study showed that sepsis induced by cecal ligation and puncture (CLP) in mice decreased the expression of Ppargc1a (encoding PGC1a), and this reduction was linked with decreased number of total mitochondria, increased in the percentage of injured mitochondria and impaired oxidative phosphorylation [194]. Pharmacological activation of PGC1a promotes defense against human rhinovirus infections (HRV) [125]. Suliman, HB and coworkers showed that nuclear accumulation of PGC-1 was a transcriptional response to lipopolysaccharide (LPS) administration in rat heart, which caused reduced mtDNA copy number and increased oxidative stress [195]. Other study showed that the *Pseudomonas aeruginosa* metabolite 2-amino acetophenone (2-AA) downregulated the expression of PGC1b and PPARy and this was associated with mitochondrial dysfunction in skeletal muscle and depletion of ATP synthesis in mice [196].

Activation of PGC1a is mediated by AMP-activated protein kinase (AMPK), which is considered a guardian of the cell metabolism and mitochondrial homeostasis [197]. Under cellular stress, AMPK phosphorylates specific enzymes such as mitofusins to control the dynamics of mitochondrial fusion and fission, autophagy-related protein 9 (ATG9) and serine/threonine-protein kinase ULK1 proteins to modulate autophagy and mitophagy, phospholipase D1 (PLD1) and thioredoxin-interacting protein (TXNIP) to control glucose uptake, acetyl-CoA carboxylases ACC1 and ACC2 of fatty-acid and sterol synthesis and lipid β-oxidation, PGC1a to regulate mitochondrial biogenesis, among others [197]. AMPK can either inhibit anabolism favoring a decrease in ATP consumption or stimulate ATP generation by catabolic mechanisms. Proper activation of AMPK-PGC1a pathway is required for antimicrobial host defense of several pathogens [198].

The PGC-1a response also included interaction with the redox-regulated nuclear respiratory factors NRF1 and NRF2. These factors mediate the coordination between nuclear and mitochondrial genomes by regulating the expression of several nuclear-encoded ETC proteins and by activating the mitochondrial transcription factor (TFAM), thus involved in the mitochondrial-encoded cytochrome C oxidase (COX) subunits and mitochondrial DNA replication [199,200]. This communication reveals an integration of multiple transcriptional regulatory pathways to sustain homeostasis in a tissue specific way [201]. During infection, studies reported altered expression of NRF1 and/or NRF2 in viral infection with DENGV [202,203], Influenza A virus [114], HIV [204,205], HSV-1 [206], EBV [109,207,208,209], SARS-CoV-2 [210] and human T-cell leukemia virus type 1 (HTLV-1) [211]. Similarly, bacterial infection was shown to modulate the expression of NRF1/NRF2, such as infection with *Staphylococcus aureus* [212,213,214] and *Escherichia coli* [215]. The expression of NRF2 was also modulated after protozoan infection with *Plasmodium* spp. [216], *Leishmania* spp. [217], *Toxoplasma gondii* [218] and *Entamoeba histolytica* [219]. *T. cruzi* infection decreased the protein expression of Nrf1 and Nrf2 in infected cardiomyocytes [220], and overexpression of Nrf2 was correlated with reduced parasitism [221].

Mitochondrial intermediate metabolites mediate epigenetic modifications, and these modifications are affected during infectious processes. For example, the mitochondrial acetyl-CoA metabolism drives histone acetylation, leading to chromatin opening and gene regulation in response to nutrient availability, metabolic reprogramming, signaling cues or stress [222]. Indeed, in vitro depletion of mtDNA caused chronic mitochondrial dysfunction, decreased enzymatic activity of histone acetyltransferases (HATs) and consequently reduced the acetylation of histones H3K9 and H3K27 [223]. A review highlighted that DNA acetylation is a crucial factor that affects the infection capacity of herpesvirus, such as human cytomegalovirus (HCMV) and Epstein–Barr virus (EBV) [224]. Bacterial infection of the human monocytic cell line THP-1 with *M. tuberculosis* infection decreased histone acetylation at the endogenous high-affinity Fc-gamma receptor (FCGR1A) promoter, which resulted in impaired response to IFN-γ [225]. *L. monocytogenes* induced deacetylation of H3K18 by mediating nuclear translocation of the deacetylase SIRT2 in vivo and in vitro [226]. Inhibition of histone deacetylase was shown to enhance mitochondrial reactive oxygen and to reduce significantly the intracellular bacterial loads of human macrophages infected with *Salmonella enterica serovar Typhimurium* and *Escherichia coli* [227]. Several metabolic pathways generate mitochondrial acetyl-CoA, including pyruvate conversion to acetyl-CoA by the pyruvate dehydrogenase complex (PDC), amino acid metabolism, conversion of acetate to acetyl-CoA by mitochondrial acyl-CoA synthetase short chain family member 1 (ACSS1), ketone body metabolism and fatty acid β-oxidation [226]. Mitochondria is also involved in the regulation of DNA and histone methylation. Mitochondria-TCA-generated α-Ketoglutarate (α-KG) is an important cofactor of the lysine demethylase 6B (KDM6B), histone demethylases (HDM) and TET DNA methylases, while fumarate, succinate and 2-hydroxyglutarate [2-HG) are inhibitors of TET [228]. Chromatin remodeling through a KDM6B-dependent demethylation of H3L27 within the IL-11 promoter was reported to facilitate the p65-NF-kB interaction, which “fine-tuned” cellular response against *Streptococcus pneumoniae* [229]. Pathogen-induced metabolic disturbances can, therefore, affect the mitochondrial metabolism, misbalance the production of mitochondrial metabolites involved in chromatin remodeling culminating in improper immune response.

In general, the proper coordination between mitochondria, metabolite generation and activation of nuclear genes PGC1a/NRF1 and NRF2 is crucial to sustain cell survival. Dysregulation in the mito-nuclear communication provoked by infectious pathogens triggers oxidative stress, mtDNA oxidation and mtDNA replication impairment. Consequently, therapeutic intervention targeting molecules involved in the mito-nuclear crosstalk might be beneficial for the treatment of infectious diseases.

## 6. Therapeutic Strategies Targeting Mitochondria to Fight Infection

As described during the review, the role of mitochondria plays a crucial role during infection and inflammation. The role of mitochondria in these biological processes makes this organelle a good promising candidate for the development of therapeutic strategies. Indeed, several studies already investigated mitochondria-targeted therapeutic strategies focusing on processes such as mitophagy, ROS production or inflammasome to improve response to pathogenic infection (Table 2).

### 6.1. Mitophagy

The mitophagy is the quality control of the mitochondria network by eliminating defective mitochondria. Infection by microorganisms causes inflammation and mitochondrial damage, such as a decrease in membrane potential, ATP production, fission of mitochondria and disrupted of mitochondrial network. Many components of the mitochondria such as mtROS, mtDNA can be perceived by the cell as inflammatory signals [100].

Controlling mitophagy could be a good strategy to limit excessive inflammation and restore homeostasis in case of infection. For example, infection with single-stranded RNA (ssRNA) viruses such as HIV was shown to inhibit mitophagy [250]. ssRNAs also activate the inflammasome, which lead to an increased activation of the inflammasome and increase mitochondrial damage. This was shown in a study in microglia, which resulted in neurotoxicity [251]. Activation of mitophagy limits mitochondrial damage and cell death caused by HIV infection [252]. Many infectious diseases cause excessive inflammatory responses, such as Chagas disease or sepsis and activation of mitophagy is a control mechanism of excessive inflammation by reducing the amount of damaged mitochondria. Some authors are modulating mitophagy in disease models. A study conducted by D’Amico and co-workers showed that Urolithin A (UA), a gut microbiome-derived natural compound, increased mitophagy and reduced pro-inflammatory cytokine production and decreased inflammation [253]. In addition, activation of autophagy by UA had beneficial effects in different chronic inflammation pathologies such as colitis in mice [230], diabetes in mice [231], cardiomyopathy in rats [232] and neuronal tissue disease models [220,233]. Additionally, in a cellular model of Parkinson’s disease, a neo-substrate of the PINK1 enzyme has been identified. Indeed, this new substrate, kinetin triphosphate (KTP), a small molecule analogous to ATP, will have a higher catalytic efficiency than its endogenous substrate ATP. This will allow in vitro in this cellular model of Parkinson’s disease to increase the enzymatic activity of PINK1 and to increase mitophagy [234].

### 6.2. Mitochondrial ROS

Another possible therapeutic strategy is to limit the generation of mtROS. In order to decrease the production of mtROS, treatments with mitochondrial antioxidants are under investigation. The mitochondria-targeted antioxidant MitoTEMPO has been tested in mice in a model of influenza A virus infection [118]. Infection by the influenza virus promoted mtROS production, but intranasal treatment with mitoTEMPO reduced the amount of mtROS in the lungs, which improved mouse survival [194]. It was also shown that human embryonic kidney cells and human alveolar epithelial cell lines treated with the mtROS-specific antioxidant mitoquinone mesylate (MitoQ) and infected with respiratory syncytial virus had lower mtROS production, which limited viral infection [235]. The antioxidant MitoQ also reduces necroinflammation in the liver of patients with chronic hepatitis C virus [236]. The use of these different mtROS-specific antioxidants in these infection models is evidence that mtROS limitation is a potential therapeutic strategy for infectious diseases.

In addition, the natural phenolic compound resveratrol is repeatedly reported as an antioxidant treatment. This natural antioxidant was shown to ameliorate inflammation by reducing the secretion and the expression of pro-inflammatory factors, such as molecules of the NF-κB pathway, TNF-α, IL6 and ROS production [240]. The role of resveratrol as strong anti-inflammatory agent on infectious disease was reported in a study that investigated acute pharyngitis in rabbit model [240] and in patients with colorectal cancer [241].

### 6.3. Inflammasome

Limiting the activation of the inflammasome could help to limit the inflammation and to improve the phenotype of the patients with chronic inflammation diseases such as Chagas disease [254], hepatitis C virus [255], HIV [256], uncontrollable inflammation such as sepsis [121,256] or the development and progression of cancer [257]. In a colitis mouse model, it was shown that MitoQ suppressed NLRP3 inflammasome activation [237]. MCC950 is a molecule that inhibits both the canonical and non-canonical inflammasome pathways, preventing its formation [258,259,260]. This molecule was shown to reduce neuroinflammation in study models of Alzheimer’s disease [238,261]. The use of MCC950 is expected to enter in a phase II clinical trial for Alzheimer’s disease and other neurological disorders such as Parkinson’s disease and motor neuron disease to assess if this molecule can decrease neuroinflammation [238]. The molecule MCC950 was also tested in bone marrow cells derived from mice infected with *Mycobacterium tuberculosis* and results showed that MCC950 blocked the NLRP3 inflammasome and reduced bacterial survival [239].

Targeting the NLRP3 inflammasome is a therapeutic strategy against diseases with significant inflammation to consider. It would even be interesting to study it in relation to the current COVID-19 epidemic [262].

### 6.4. Mitochondrial Dynamics

The mitochondrial dynamics sustain cell homeostasis and the mitochondrial network. The fusion mechanism is important for ATP production to meet the energy needs of the cell. While the fission mechanism is important in the generation of new mitochondria and the quality control of these. As described in previous sections, the pathogens can manipulate mitochondrial dynamics to favor their intracellular replication. Many of them will cause mitochondrial fission. Some studies described molecules that modulate the mitochondrial fission process. Indeed, the molecule P110 is a specific inhibitor of the interaction of DRP1/FIS1 [243]. The P110 was shown to mediate mitochondria dysfunction by effectively inhibiting the interaction DRP1/FIS1 interaction in in vitro and in vivo models of septic cardiomyopathy [243]. Moreover, P110 diminished neurotoxicity by inhibiting aberrant mitochondrial fission in neuron cell lines model of Parkinson’s disease [244]. Additional molecules are known to inhibit DRP1-mediated mitochondrial fission, such as melatonin and mdivi-1. Inhibition of fission by melatonin and mdivi-1 was shown to limit mtROS production in diabetic mice by a mechanism that involves the SIRT1-PGC1α pathway [242], which could also have beneficial impact on infectious disease models.

### 6.5. Artificial Mitochondrial Transfer

A newly studied therapeutic strategy is the cell-to-cell mitochondria transfer, especially between cells of different types. This cell-to-cell transfer of mitochondria was first shown between human mesenchymal cells and mitochondria-deficient cancer cells. The transferred mitochondria re-established a mitochondrial network in the mitochondria-deficient cancer cells [250]. In inflammation, mitochondrial damage accumulates and amplifies the inflammatory response, so it might be thought that transferring healthy mitochondria would regulate the inflammation. A team demonstrated that the regulation of T cells by mitochondrial transfer was possible and that it could attenuate inflammation [246]. Indeed, they used a technique called mitoception [245] which allows the transfer of mitochondria from human mesenchymal stem cells to T cells and that this transfer reprogramed the cells to promote the induction of Tregs, thus regulating the inflammation by activating on the expression of FOXP3, IL2RA, CTLA4 et TGFβ1 factors those that will increase CD25+ FoxP3+ population. [246]. Autologous mitochondria transfer is currently being studied extensively in the reduction of ischemia after reperfusion. Two in vivo studies reported that autologous mitochondria transfer in rabbit protected the heart from ischemia-reperfusion injury [247,263]. Mitochondria transfer also resulted in the reduction of ischemic reperfusion injury in rat’s liver [248] and brain [249]. Further studies are necessary to investigate if mitochondria transfer has potential to be a therapeutic strategy to improve mitochondria function in chronic inflammatory and/or infectious diseases.

## 7. Conclusions

In this review, we explored how the major mechanisms involved in the mitochondrial bioenergetics and dynamics are modulated by inflammation and pathogen infection. We also addressed the therapeutic strategies that are underway. The host cell mitochondria manipulation by pathogens is an effective evolutionary mechanism that allows the pathogens to effectively infect, survive and proliferate inside the potentially harsh intracellular environment.

It has become clear that chronic inflammatory and infectious diseases affect mitochondria by either disrupting the mitochondria networks or by promoting excessive, uncontrolled production of reactive oxygen species. Mitochondria dysfunction and oxidative stress are mediators of improper immune response, tissue disruption and organ failure and thus are perfect targets for disease control. With the ever-increasing demand of new therapeutic strategies, the mitochondria’s role in disease needs to be further explored as it has shown to have great potential for the development of refined therapy for viral, bacterial and protozoan infections and chronic inflammatory diseases.

## Figures and Tables

**Figure 1 ijms-22-11338-f001:**
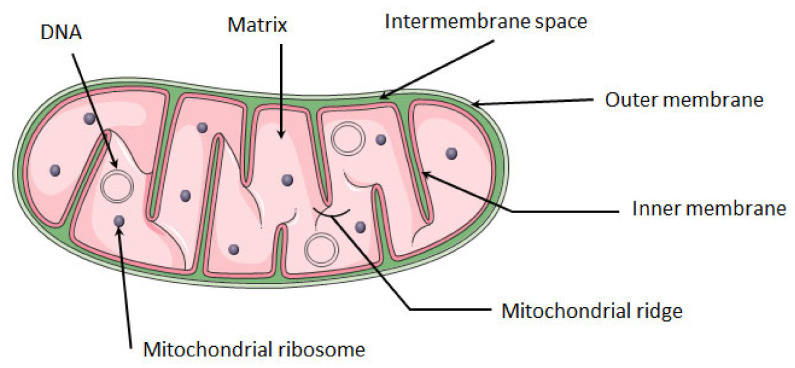
**Structure of mitochondria.** The mitochondrion is composed of a double membrane: the inner membrane and the outer membrane. Between these membranes is the intra-membrane space. The inner membrane forms invaginations called ridges where the OXPHOS complexes are located. The mitochondrial matrix contains several copies of mitochondrial circular DNA and ribosomes. Mitochondria image adapted from Smart. Available online: https://smart.servier.com (accessed on 10 August 2021).

**Figure 2 ijms-22-11338-f002:**
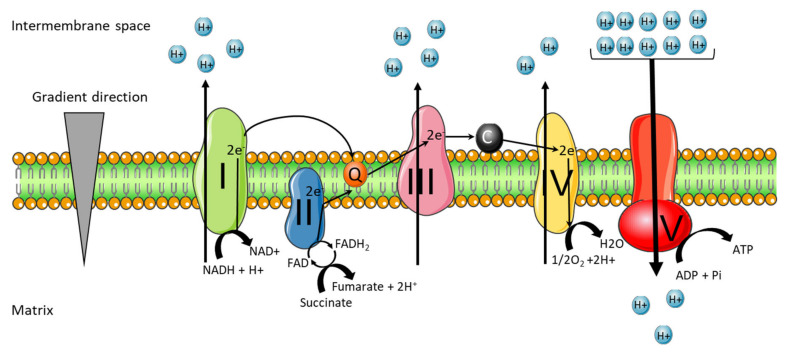
**Mitochondrial respiratory chain complex.** The mitochondrial respiratory chain is located at the inner membrane of the mitochondria. Composed of four complexes and two coenzymes, it allows the production of ATP through oxidative phosphorylation. Complex I (NADH: coenzyme Q oxidoreductase) and II (succinate dehydrogenase) will each transfer two electrons to the quinone coenzyme Q10 (CoQ10). The two electrons transferred from complex I come from the oxidation of NADH, and those from complex II come from oxidation of succinate to fumarate. CoQ10 will allow the transfer of electrons to complex III (CoQ10-cytochrome C oxidoreductase). The complex III will then pass these electrons to the cytochrome C, which makes the link with the complex IV (cytochrome C oxidase). The complex IV reduces O_2_ in a H_2_O molecule. Complexes I, III and IV are proton pumps, which allow the passage of protons from the matrix to the intermembrane space, in the opposite direction of the gradient. Complexes I and III allow the passage of four protons and complex IV of two protons. Once the intermembrane space is enriched with protons, the last complex of the chain, ATP synthase will allow the passage of protons in the direction of the gradient. This proton flow will allow the synthesis of ATP from ADP.

**Figure 3 ijms-22-11338-f003:**
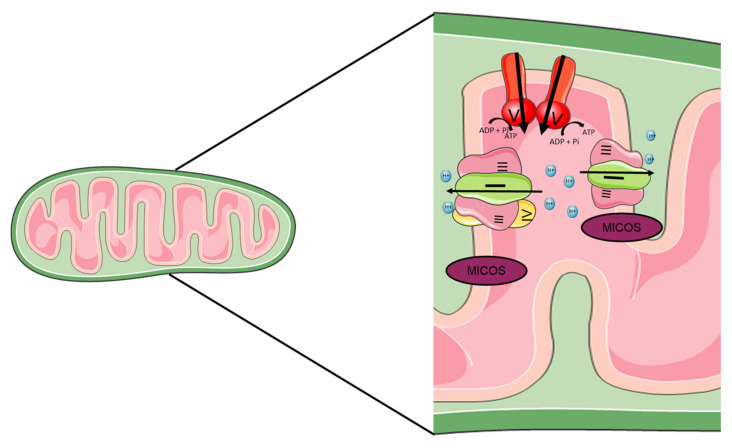
**Structure of mitochondrial supercomplexes.** The structure of the mitochondrial ridges is very important in the function of mitochondria and in particular in the stability of the super complex. Several proteins are involved in the stability of these ridges, the ATP synthase which will be at the top and the mitochondrial contact site and cristae organizing system (MICOS) which will be at the bottom. Between the two will be the super complexes. The two main super-complexes found in mammals I + III2 + IV1–4 and I + III2 are shown.

**Figure 4 ijms-22-11338-f004:**
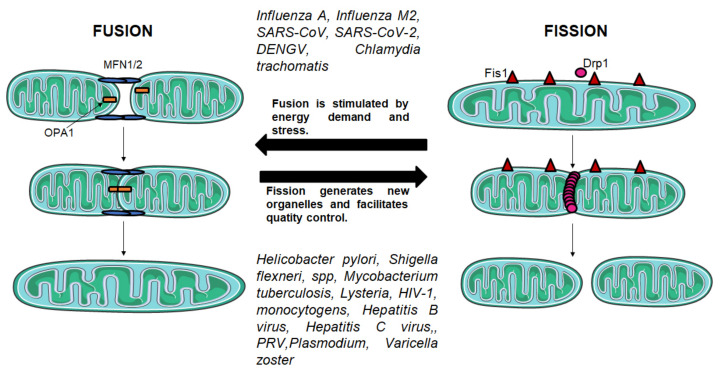
**Fusion and fission mechanisms.** Fusion and fission mechanisms are important in the regulation of mitochondrial morphology and function. Fission is mediated by DRP1, a GTPase which is recruited from the cytosol by FIS1 to form spirals around mitochondria that constrict both inner and outer membranes. Fusion between mitochondrial outer membranes is mediated by MFN1 and MFN2, whereas fusion between mitochondrial inner membranes is mediated by OPA1.

**Figure 5 ijms-22-11338-f005:**
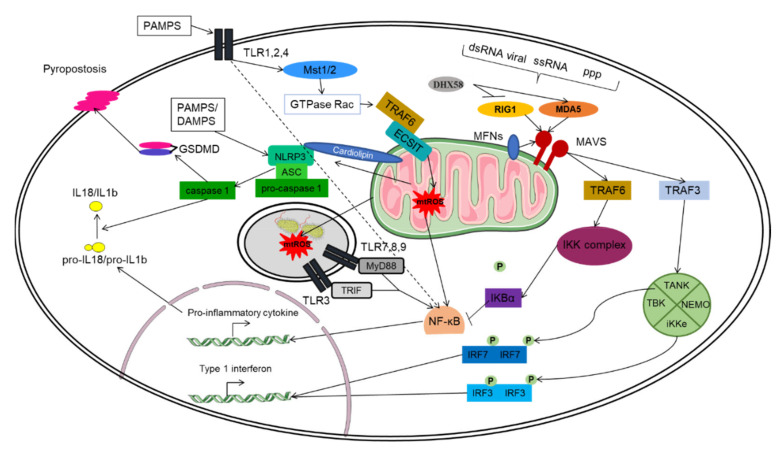
**The mitochondria at the center of the RLR, NLR and TLR pathways.** The mitochondrion plays an important role in the innate immune response, as it participates in the main pathways involved in the immune response: the TLR, RLR and NLR pathways. The TLR pathway is involved in the antimicrobial response. PAMPs will activate the TLRs. The membrane-spanning TLRs (TLR1, TLR2, TLR4) and endosome-bonded TLRs (TLR3, TLR7, TLR8 and TLR9) are activated and trigger the expression of pro-inflammatory cytokines via the NF-kB pathway. The TLRs activate the kinases Mst1 and Mst2, which activate the GTPase Rac allowing the assembly of the TRAF6-ECSIT complex. The TRAF6 translocates to the mitochondria and binds to ECSIT, forming the TRAF6-ECSIT complex, which lead to an increase in mitochondrial ROS (mtROS) production as well as the proximity of the mitochondria and the phagosome. The bacteria in the phagosome will be destroyed by the action of mtROS. The RLR pathway is involved in antiviral response. Different PAMPs such as dsRNA, ssRNA and 5′-triphosphate (ppp) will activate two cytoplasmic PRRs, RIG-1 and MDA5. These PRR are regulated by DHX58, where DHX58 positively regulates MDA5 and negatively regulates RIG-1. After the interaction with DHX58/RIG1/MDA5, MAVS is activated by recruiting different proteins, such as TRAF3 and TRAF6. The interaction between MAVS and the MFN1 and MFN2 fusion proteins (MFNs) are essential for RLR pathway signaling. TRAF3 will activate the TANK/NEMO/iKKe/TBK complex, which will phosphorylate IRF7 and IRF3 dimers, allowing their translocation into the nucleus and their binding to ISRE sequences at the promoters of type I IFN-regulated genes, while TRAF6 will activate the IKK complex. This complex activates NF-ĸB by phosphorylating its inhibitor IKBα. Once activated, NF-ĸB translocates to the nucleus to induce the expression of pro-inflammatory genes. The NLR pathway is involved in a mechanism for fighting infection and cellular damage. The main complex of this pathway is the NLRP3 inflammasome. The inflammasome is a multiprotein complex containing ASC, pro-caspase 1 and NLRP3. The activation of NLRP3 inflammasome is done both on transcriptional and post translational levels. The first is related to TLR activation and production of pro-IL1b and pro-IL18. The second signal is based on the detection of PAMPs and DAMPs that leads to NLRP3 oligomerization and interaction with ASC. Then, ASC recruits pro-caspase 1 via the CARD domain. Once caspase 1 is activated, it cleaves the pro-IL1b and pro-IL18 which will allow the release of these inflammatory cytokines in the extracellular milieu. In addition, caspase 1 cleaves Gsdmd, which will allow pore creation at the plasma membrane and pyroptosis. In addition, MAVS is also necessary for the recruitment of NLRP3 in mitochondrial membranes. Cardiolipin and mtROS promote the activation of NLRP3 inflammasome.

**Table 1 ijms-22-11338-t001:** Effect caused by bacteria, protozoan and viruses on mitochondria bioenergetics and dynamics.

Organism	Pathogen	Effect on Mitochondria
Virus	PRV	Fission [107]
	DENGV	Fusion [108]
	EBV	Reduced mitochondria content [109,110] Reduced Δψm, reduced ATP [111]
		Increased ROS and MMPT opening [111]
	Hepatitis B virus	Fission, perinuclear distribution [112]
	Hepatitis C virus	Fission, perinuclear distribution [113]
	HIV-1	Increased OXPHOS [114,115]
		Decreased Δψm [116]
		Fission, perinuclear distribution [116,117]
	Influenza A	Fusion and increased ROS [118]
	Influenza M2	Fusion and increased ROS [119]
	SARS-CoV	Fusion [120]
	SARS-CoV-2	Fusion [121,122,123]
		Increased ROS [124]
	HRV	Decreased ROS [125]
	HPV	Reduced cristae [126] Reduced Δψm [127]
		Increased ROS [126,128]
	Varicella zoster	Fission [129]
		Warburg effect [129,130,131]
Bacteria	*Listeria monocytogenes*	Fission [132]
		DRP1-independent fission [133,134]
	*Legionella pneumophila*	Reduced ATP, reduced oxygen consumption [135]
	*Chlamydia trachomatis*	Fusion [136]
		ROS production [137,138]
	*Helicobacter pylori*	Fission [139]
	*Shigella flexneri*	Fission [140,141]
	*Mycobacterium tuberculosis*	Fission, perinuclear distribution [137,138]
		Reduced Δψm [142,143]
		Increased Δψm and ATP [143]
Protozoa	*Trypanosoma cruzi*	Decreased OXPHOS activity [144,145]
		Reduced activity of GPx, GSH and MnSOD [146,147]
		Increased ROS [148,149,150]
		Increased oxidative stress [151,152]
		Increased basal respiration, proton leak and ATP production [153]
	*Plasmodium* spp.	Impaired Δψm, increased ROS [154]
		Fission and apoptosis [155,156,157]
	*Toxoplasma gondii*	Fission [158]
		Fusion [159]
		Increased ROS [158]
		Changed metabolism [160] OXPHOS [161]
		Decreased OXPHOS proteins [158]
	*Leishmania* spp.	Impaired Δψm [162]
		Metabolic shift [163]

**Table 2 ijms-22-11338-t002:** Therapeutic strategies targeting mitochondria in inflammatory and infectious disease models.

Mechanism	Compound	Infection or Inflammatory Disease Model
Induce Mitophagy	Urolithin A	Colitis in mice [230]
		Diabetes in mice [231]
		Cardiomyopathy in rats [232]
		Neuronal tissue disease models [220,233]
	Kinetin triphosphate	Cellular model of Parkinson’s disease [234]
Limit mtROS	MitoTEMPO	Influenza A virus infection in mice [118]
	Mitoquinone mesylate (MitoQ)	Cell lines infected with respiratory syncytial virus [235]
		Patients with chronic hepatitis C virus [236]
Inhibit NLRP3 inflammasome	Mitoquinone mesylate (MitoQ)	Colitis in mice [237]
	MCC950	Class II clinical trials for Alzheimer’s disease, Parkinson’s disease and motor neuron disease [238]
		Mice infected with mycobacterium tuberculosis bacteria [239]
Reduce the secretion and expression of inflammatory factors: inhibit NK-κb pathways, TNF-α, IL6 and ROS	Resveratrol	Rabbit models of acute pharyngitis [240]
		In colorectal cancer patients [241]
Inhibit mitochondrial fission: inhibit Drp1 and limit mtROS	Melatonin and mdivi-1	Diabetic cardiomyopathy in diabetic mice [242]
	P110	In vitro in LPS-treated H9C2 cardiomyocytes and in vivo in septic cardiomyopathy mice models [243]
		In vitro model of Parkinson’s disease [244]
Artificial mitochondrial transfer Mitoception [245]		Transfer mitochondria from human mesenchymal stem cells to T cells, and that this transfer would reprogram the cells to promote the induction of tregs [246]
		Rabbit model of ischemia-reperfusion injury of the heart [247]
		Rats models of ischemia-reperfusion injury of the liver [248]
		Rats models of ischemic stress in the brain [249]

## Data Availability

Not applicable.

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
