# Peer review of "Mitochondria as a Cellular Hub in Infection and Inflammation"

_ijms, 2021, doi:10.3390/ijms222111338_

Round 1

Reviewer 1 Report

I would thank the authors for the changes they have made to the manuscript.

Reviewer 2 Report

No new comments are required from my side

This manuscript is a resubmission of an earlier submission. The following is a list of the peer review reports and author responses from that submission.

Round 1

Reviewer 1 Report

In this review Andrieux and collaborators have summarised a very broad set of elements to collate a very useful information about the role of mitochondria in infection and inflammation. The started from the origen of the mitochondria, they explained its structure and function in very good manner, followed by the adaptation of mitochondria to the host cell, before walking around the role of mitochondria in the infection of a good pool of pathogens covering virus, bacteria and parasites. They continued with the process of inflammation and pointed some therapeutic approaches based in potential targets. 

The review is well written and clear to follow. So, it meets the purpose of a review paper by providing to researchers a value source of information to understand the play of mitochondria in the processes described. This review will also facilitate other researcher to access to precise information of a range of organisms and processes due to the extensive and appropriate references given. 

The two weakness I have spotted, and which improvement would increase the quality of the manuscript, are:

  1. The therapeutic strategy information is quite vague and speculative, rather than precise - this full section should be amended or removed. Alternatively unify in one single section without the headlines. 
  2. The conclusions section is pretty short and it doesn't bring a deep conclusion style from all the information provided before. To me is more like an abstract.

Reviewer 2 Report

In the review “Mitochondria as a cellular hub in infection and inflammation”, the authors widely describe the origin of mitochondria and focus on their involvement on the initiation and progression of infective diseases.

Overall, the review is well written, but there are few concerns that need to be addressed.

First, regarding the division in paragraph, there are for sure typos on the numbering, since the paragraphs’ numbers are either missing or a repetition of 1. Of course, this leaves a lot of confusion in the reader.

Moreover, the main focus of the review is about mitochondria in inflammation, therefore I suggest reconsidering the paragraph division having an introduction mainly focused on mitochondrial origin, mitochondrial metabolic functions and mitochondrial dynamics, then a second paragraph focused on mitochondria and infections, then the possible therapeutic strategies. In the actual state, the manuscript is difficult to read, not because of the contents but due to the confusion generated by the paragraphs’. In the same way: why mitophagy is named only at the end of the manuscript and it is not described in the mitochondrial dynamics? Or what is the purpose of describing all the mitochondrial respiratory chain since it will not be further discussed in an infectious disease scenario? These are just few questions to help the authors trimming the excess of information and define a stronger outline.

The literature revision conducted by the authors is admirable, but it misses the contribution to the “mitochondria & inflammation” field of publications from different research groups, such as Prof. Tait, Prof. Pinton, Prof. Geny, Prof. Green, and Prof. Giorgi, just to cite a few.

Lastly, there are several grammar typos, therefore it would be useful the revision by a native English speaker.